# Visual Interpretation of Convolutional Neural Network Predictions in Classifying Medical Image Modalities

**DOI:** 10.3390/diagnostics9020038

**Published:** 2019-04-03

**Authors:** Incheol Kim, Sivaramakrishnan Rajaraman, Sameer Antani

**Affiliations:** Lister Hill National Center for Biomedical Communications, National Library of Medicine, 8600 Rockville Pike, Bethesda, MD 20894, USA; sivaramakrishnan.rajaraman@nih.gov (S.R.); santani@mail.nih.gov (S.A.)

**Keywords:** class-selective relevance mapping, convolutional neural network, modality classification, visual localization, discriminative region of interest

## Abstract

Deep learning (DL) methods are increasingly being applied for developing reliable computer-aided detection (CADe), diagnosis (CADx), and information retrieval algorithms. However, challenges in interpreting and explaining the learned behavior of the DL models hinders their adoption and use in real-world systems. In this study, we propose a novel method called “Class-selective Relevance Mapping” (CRM) for localizing and visualizing discriminative regions of interest (ROI) within a medical image. Such visualizations offer improved explanation of the convolutional neural network (CNN)-based DL model predictions. We demonstrate CRM effectiveness in classifying medical imaging modalities toward automatically labeling them for visual information retrieval applications. The CRM is based on linear sum of incremental mean squared errors (MSE) calculated at the output layer of the CNN model. It measures both positive and negative contributions of each spatial element in the feature maps produced from the last convolution layer leading to correct classification of an input image. A series of experiments on a “multi-modality” CNN model designed for classifying seven different types of image modalities shows that the proposed method is significantly better in detecting and localizing the discriminative ROIs than other state of the art class-activation methods. Further, to visualize its effectiveness we generate “class-specific” ROI maps by averaging the CRM scores of images in each modality class, and characterize the visual explanation through their different size, shape, and location for our multi-modality CNN model that achieved over 98% performance on a dataset constructed from publicly available images.

## 1. Introduction

Medical images serve as an indispensable source of information in clinical decision-making [1]. They are generated through different imaging modalities in hospitals and clinics, including magnetic resonance imaging (MRI), computed tomography (CT), positron emission tomography (PET), ultrasound, chest radiography, etc. Large medical image repositories have become an essential part of the analytical content in biomedical research and related publications [2], and they provide opportunities for developing more reliable computer-aided detection (CADe) and diagnosis (CADx), and information retrieval algorithms. For these reasons, automated medical image modality classification and retrieval techniques have gained immense significance in clinical and biomedical research and visual information retrieval fields in recent years.

A number of evaluation campaigns like Kaggle (https://www.kaggle.com) and Image Cross Language Evaluation Forum (ImageCLEF) provide medical image collections, annotated toward several evaluation challenges including modality classification, anomaly detection, segmentation, dimensionality reduction, visual question answering, image captioning, and compound figure separation [3]. Literature studies reveal that most of the recent approaches for modality classification are machine learning-based and rely on handcrafted textual or visual features such as a bag of colors [4], bag of words [5], 2D color feature [6] including color gradient orientations, and local pattern distributions [7]. However, such handcrafted features often cannot capture variations in visual representation and the modality of an image because of intra-class variability in visual appearance and inter- and intra-observer variability in manual annotation processes. In addition, the feature extraction process is computationally expensive and demands expertise in developing algorithms, requiring extensive labeling and accounting for the limited visibility and variability in morphology and position of the region of interest (ROI) for modality detection [8].

Data-driven approaches like deep learning (DL) are presented as a breakthrough alternative to extract discriminative image features for retrieving various imaging modalities with minimal human intervention. Convolutional Neural Network (CNN) is the most commonly used DL architecture in many image analyses and computer vision-related tasks. It automatically learns a hierarchical representation of visual features from the input image pixels [9] through mechanisms of multiple (deep) layers of receptive fields, weight sharing, and pooling [10], and it has shown superior performance compared to other conventional machine learning methods [11]. Yu et al. [12] classified the ImageCLEF2013 medical modality dataset using customized CNNs with an accuracy of 74.9%. Koitka and Friedrich [13] used a pretrained residual network (ResNet) as a fixed feature extractor to train a support vector machine (SVM) classifier toward obtaining a classification accuracy of 85.4% with the ImageCLEF2016 modality classification dataset. Combinations of customized and pretrained CNN models were also introduced to classify the ImageCLEF2016 modality dataset or ImageCLEF2016 sub-figure datasets with an accuracy of 82.5% [14], 87.4% [15], and 86.58% [16], respectively.

While the recent literature discusses various effective and promising DL models for modality classification, none of these attempted to visualize, understand, or interpret the internal representations and prediction results of their DL models. A poorly understood model behavior adversely impacts decision-making, introducing inaccuracies in visual information retrieval and visual question-answering applications. Therefore, the need for exploratory studies for interpreting and explaining the learned behavior of DL models toward medical modality classification is significant and is the rationale behind this study. There have been some advances in recent years, though made in other computer vision tasks, to gain a better insight into DL models. Zeiler and Fergus [17] proposed a multi-layered deconvolutional network model to project the feature activations back to the input pixel space for visualizing input stimuli that excite individual feature maps at any layer in the model. Mahendran and Vedaldi [18] conducted a visual analysis of internal representations at different layers by inverting them. However, these approaches just show how an input image is represented by the deep features in the DL model without addressing what information or which region in the image is most important and class-discriminative for a correct prediction.

In this paper, we propose a novel method of localizing and visualizing a region of interest (ROI) within a medical image that is considered to be the most discriminatively important in order to explain the predictions of the CNN-based DL models toward the challenge of classifying medical imaging modalities. The concept of “Class-selective Relevance Mapping” (CRM) is newly introduced. It measures the importance of the activation at a spatial location (x,y) in the feature maps produced from the last convolution layer that resulted in correct classification of an input image. The proposed CRM is based on a linear sum of incremental mean squared error (MSE) between target and actual outputs calculated from all nodes in the output layer of the CNN model. A spatial element at location (x,y) in the feature maps from the last convolution layer can be considered significant in producing the correct prediction if removing that element results in a significant increase in the MSE at the final output layer. The resulting “class-selective relevance map” in which each component quantitatively indicates the importance of its corresponding spatial element in the feature maps is then projected back to the input image to locate and highlight the ROI. It is worth noting that our CRM considers both positive and negative contributions of each element in the feature maps to the model prediction (i.e., a contribution to increasing the output score for the desired class and decreasing the ones for other classes) in calculating their importance. Therefore, the resulting ROI is intrinsically class-discriminative.

There have been a number of recent works that also evaluate the importance of the feature maps to produce a visual explanation of the CNN prediction. Zhou et al. [19] proposed a technique called “class activation mapping” (CAM) for identifying a region in the input image that is important to a particular category. Basically, CAM requires a specific type of CNN architecture (a global average pooling (GAP) and a single dense layer on the top of the last convolution layer), and is derived from a prediction score at one specific output node, representing the image class to which a given input image belongs. Another visualization method—“gradient-weighted class activation mapping” (Grad-CAM) [20]—has also been suggested as a generalization version of CAM, as it can be applied to any CNN-based DL models without any modification of their architectures. Unlike our method considering prediction scores from all output nodes, these two visualization methods rely on only one prediction score from a particular output node.

In our experiments, we first designed and developed a “multi-modality” CNN model using the pretrained VGG16 model [21]. This CNN model was trained to classify seven different modalities of medical images: (1) abdomen CT, (2) brain MRI, (3) cardiac abdomen ultrasound, (4) chest X-ray, (5) fluorescence microscopy, (6) retinal fundoscopy, and (7) statistical graphs. Next, we analyzed its behavior using the proposed CRM by localizing and visualizing discriminative ROIs in the images for each modality class. We also compared the localization performance of our CRM with that of aforementioned existing visualization methods.

## 2. Materials and Methods

### 2.1. Data Collection and Preprocessing

The dataset for training and testing of our DL model was pooled from multiple resources including the Open Access Biomedical Image Search engine (Open-i^®^, Available at: https://openi.nlm.nih.gov) at the U.S. National Library of Medicine (NLM), the ImageCLEF2013 modality classification challenge [8], and the World Wide Web. We also collected additional images for abdominal CT from the Cancer Imaging Archive [22] and for retinal fundoscopy from the publicly available MESSIDOR (Methods to evaluate segmentation and indexing techniques in the field of retinal ophthalmology) [23] and Digital Retinal Images for Vessel Extraction (DRIVE) [24] databases, respectively. Some details about images in our resulting dataset are shown in Table 1, and a sample image of each modality category is also shown in Figure 1.

We augmented our dataset by introducing class-specific perturbations during a training phase to make our DL model robust and to reduce bias and generalization errors. These perturbations generated new samples for training without actually altering visual characteristics of the images. Data were augmented with rotations and horizontal and vertical shifts in the ranges [−5, 5] and [−3, 3], respectively. Images were then resampled to 300 × 300-pixel resolutions and normalized for mean to assist model convergence.

### 2.2. Convolutional Neural Network (CNN) Configuration

Our “multi-modality” CNN model was built based on the pretrained VGG16 network. We first truncated the VGG16 at the deepest convolutional layer. A convolutional layer with 512 filters, each with a 3 × 3 spatial dimension, followed by a global average pooling (GAP) and a dense layer was then added on top of the truncated model, as shown in Figure 2. Our model was trained with small weight updates to self-discover/learn the hierarchical feature representations in modality images to result in end-to-end feature extraction and classification.

A randomized grid search was performed to find the optimal values for the hyper parameters [25]. The search ranges for the learning rate, momentum, and L2-regularization were set as [1e−5, 1e−1], [0.8, 0.99], and [1e−10, 1e−1], respectively. We also used a mini-batch size of 10 resulting in 3500 iterations per epoch and “Adam” optimizer. Our “multi-modality” CNN model was implemented based on Tensorflow [26], a very well-known open source library developed by the Google Brain team, Keras [27], a simple and high-level model definition interface, and NVIDIA’s CUDA toolkit (Available at: https://developer.nvidia.com/cuda-zone) and CuDNN for GPU-acceleration. The training was performed on the NVIDIA-DGX1 facility having Tesla V-100 GPUs, available at NLM.

## 3. Class-Selective Relevance Map (CRM)

First, we briefly describe the existing methods of localizing and highlighting important ROI for a particular category within an input image developed to visually explain the CNN prediction. Next, we provide the details of our proposed visual localization method and compare it with these existing methods through a sequence of experiments.

### 3.1. Class Activation Map (CAM)

Essentially, CAM [19] was generated from a specific CNN architecture where global average pooled convolutional feature maps were fed into the fully connected final output layer. Let Fk∈ ℝu×v represent the global average pooling that spatially averages the *k*-th feature map of width *u* and height *v* produced from the last convolution layer, and wkc be the weight connecting the *k*-th feature map to an output node corresponding to the class *c*. Then, for the class *c*, a prediction score (the input to the softmax or sigmoid function) at the output layer, Sc can be expressed as a weighted sum of the global average pooling.
(1)Sc=∑kwkcFk=∑kwkc∑x,yfk(x,y )=∑x,y∑kwkcfk(x,y),
where fk(x,y) denotes the activation at the spatial element (*x*,*y*) in the *k*-th feature map.

CAM for class *c*, Mc∈ ℝu×v was defined as a weighted sum of activation at spatial element (*x*,*y*) from all feature maps.
(2)Mc(x,y)=∑kwkcfk(x,y).

Since a prediction score $S_{c}$ was a linear sum of all elements in $M_{c}\left(x,y\right)$, CAM reflected the importance of the activation at each spatial element (*x*,*y*) for the classification of an input image to class *c*. Finally, by simply up-sampling the CAM to the size of the input image, we located the ROI within the image most important to the particular category.

### 3.2. Gradient-Weighted Class Activation Map (Grad-CAM)

Grad-CAM [20] used the gradient information of a target class flowing back into the last convolutional layer to generate visual explanations from any CNN-based DL models. Unlike the aforementioned CAM, this method did not require a specific CNN architecture (GAP and a single dense layer on the top of the last convolution layer), thereby avoiding any modification of the existing CNN architecture. Grad-CAM for the class *c* was also defined as a weighted sum of all feature maps resulting from the last convolution layer in the CNN. In addition, ReLU function was applied to remove a potential influence from negative weights on the class of interest, considering that the spatial elements in the feature maps associated with the negative weights were likely to belong to other categories in the image.
(3)$$Grad\_M_{c}\left(x,y\right)=ReLU\left(\sum_{k}\alpha_{k}^{c}f_{k}\left(x,y\right)\right).$$

Here, $\alpha_{k}^{c}$ is the weight obtained by computing the gradient of a prediction score, $S_{c}$ with respect to the *k*-th feature map:(4)$$\alpha_{k}^{c}=\sum_{x,y}\frac{\partial S_{c}}{\partial f_{k}\left(x,y\right)}.$$

According to Equations (1) and (4), this $\alpha_{k}^{c}$ was precisely the same as $w_{k}^{c}$ in the CAM for CAM-applicable CNN architectures. Therefore, applying ReLU to exclude the spatial elements with a negative weight was the only difference between Gard-CAM and CAM in such cases.

### 3.3. Class-Selective Relevance Map (CRM)

Originally, the concept of relevance [28] was introduced to measure the importance of a hidden node in a multi-layered neural network for producing the appropriate output correctly. The relevance of the *m*-th hidden node was defined by the incremental MSE at the output layer computed without that node.
(5)$$R_{m}=\sum_{p}\sum_{c}{\left(t_{c}^{p}-o_{c}^{p}\left(m\right)\right)}^{2}\;-\;\sum_{p}\sum_{c}{\left(t_{c}^{p}-o_{c}^{p}\right)}^{2}.$$

Here, $o_{c}^{p}$ and $o_{c}^{p}\left(m\right)$ denote the actual output (output of an activation function such as softmax or sigmoid) at the output node representing the class *c* computed with and without the *m*-th hidden node, and $t_{c}^{p}$ denotes the corresponding target value, respectively, upon presentation of an input *p*. Accordingly, a hidden node having a large relevance value could be considered as playing a very important role in the learning and classification process, since removing that node resulted in a significant increase in the MSE. This idea has been successfully applied in various machine learning problems related to network pruning, speed-up of learning, etc. [29,30].

In this study, we proposed a novel method based on the concept of such class-selective relevance to measure the importance of the activation of the feature maps at the last convolution layer of a CNN model at the spatial location (*x*,*y*), thereby reliably locating the most discriminative ROI in the input image. We first calculated a prediction score, $S_{c}$ at each node *c* in the output layer using Equation (1) and another prediction score, $S_{c}\left(l,\;m\right)$ after removing a spatial element, (*l*,*m*) in the feature maps from the last convolutional layer.
(6)$$S_{c}\left(l,\;m\right)=\sum_{x,y\neq l,m}\sum_{k}w_{k}^{c}f_{k}\left(x,y\right).$$

The proposed class-selective relevance mapping (CRM), R(l,m) ∈ ℝu×v was then defined as a linear sum of incremental MSE between Sc and Sc(l, m) calculated from all nodes in the output layer of the CNN-based DL model.
(7)R(l,m)=∑c=1N{(Sc−Sc(l,m)}2.

Figure 3 illustrates a conceptual workflow for calculating the CRM score from a VGG16-based DL model, and it is simplified for purposes of reader understanding to the case of a two-class problem. Deep learning is basically a discriminative learning process. Thus, an important element in the feature maps from the last convolution layer would make not only a positive contribution to increase the prediction score at the output node representing the desired class but also a negative contribution to decrease the prediction score at the remaining output nodes, thereby maximizing the gap between these prediction scores. Since our CRM was based on the incremental MSE calculated from all output nodes, the resulting ROI was expected to be class-discriminative. In contrast, the aforementioned two existing methods, viz., CAM and Grad-CAM, relied on only the prediction score from the output node for a particular class to which a given input image belonged.

In addition, we defined a “class-level” ROI that represented a region most frequently highlighted as important for the correct prediction of images all belonging to a particular class. It can be generated by averaging the CRM score of the same class of images.
(8)$$avg\_R\left(l,m\right)=\sum_{p\;\in \;c}R^{p}\left(l,m\right).$$

Here, Rp(l,m) ∈ ℝu×v denotes the CRM for the *p*-th image in class *c*. Using this “class-level” ROI, we can see the “inter-class” difference and “intra-class” similarity of ROIs between images, and we can visually explain the prediction results from our “multi-modality” CNN model.

## 4. Results and Discussion

In our experiments, we first evaluated the performance of our “multi-modality” CNN model implemented to classify seven different modalities of medical images. Next, we analyzed its behavior in modality classification by highlighting a discriminative ROI of images in each modality class located using the proposed CRM. We also demonstrated the effectiveness of our CRM by comparing its localization performance with other existing visualization methods.

### 4.1. Performance Evaluation

A VGG16-based CNN model was evaluated in terms of the performance metrics including accuracy, area under the receiver operating characteristic (ROC) curve (AUC), precision, recall, F1-score, and Matthews correlation coefficient (MCC), using an independently developed testing dataset mentioned in Table 1. It was quite difficult to compare the performance of our CNN model with those reported from other literature, owing to difference in data collections and the use of data augmentation for balancing data distribution. However, our model yielded a remarkably good performance with respect to all metrics (overall above 98%) in classifying the image modalities for the dataset under study, as can be seen from Table 2. Such high performance was expected to minimize any negative impact from a wrong prediction or a low prediction score on a precise and reliable comparison between the visual localization methods detailed in the previous section.

### 4.2. Localization Evaluation

Now, we apply the proposed CRM to identify a discriminative ROI of images in each modality class for characterizing the behavior of our CNN model. Figure 4 shows several examples of heatmap images reflecting the CRM score for a given input image. These heatmap images were generated by normalizing the CRM score to the range [0, 255] and displaying the value above 20% of the max score of the CRM.

We can first see that our CRM consistently localized and highlighed a specific area within the images as the discriminative ROI for those in the classes of (a) abdomen CT, (b) brain MRI, (c) cardiac abdomen ultrasound, (d) chest X-ray, and (f) retinal fundoscopy where images are found to have a high level of “intra-class” similarity of shapes or patterns. In particular, the heatmap examples for the retinal fundus images show that the optic disc area was successfully identified as the most important ROI (red-colored) by the proposed CRM even though its location was very different (opposite side) among images. Accordingly, our “multi-modality” CNN model was analyzed through the CRM process such that it mainly focused on a region of an input image, which was also common in shape or pattern among other images within the same class but distinct or different from those in other classes, as important and discriminative for its correct modality classification.

On the other hand, we can also observe from the heatmap examples shown in Figure 4e that our CNN model behaved as an object detector for images from the class of fluorescence microscopy. Here, all images were shown to have a very low level of “intra-class” similarity (randomly shaped) of objects. The majority of images in the statistical graph class were found to have *X* and *Y* axes (except in Venn diagram images) with a certain and consistent statistical data graph pattern (though it looked quite different among images). In such case, our CRM highlighted both *X* and *Y* axes and the statistical data graph, as shown in Figure 4f.

Next, we evaluated the localization performance of the proposed CRM by comparing it with the existing localization methods: CAM and Grad-CAM. As mentioned in the previous section, these two methods were both based on a weighted sum of feature maps from the last convolution layer, and the Grad-CAM employed the ReLU function to only consider features having a positive weight value. Figure 5 shows several examples of the heatmaps generated using each localization method. It can be first seen that the heatmaps resulting from the CAM and Grad-CAM processes were virtually the same. Table 3 also quantitatively supports this observation by showing that the average number and ratio of pixels in images from each modality class highlighted by CAM or Grad-CAM were almost the same. Our analysis revealed that the effect of ReLU function in Grad-CAM was almost completely offset by only using the mapping score value above 20% of the maximum score of the CAM and CRM to generate their heatmaps. 

Figure 5 also demonstrates that our CRM had a significant noise removal effect on heatmaps, as it consistently highlighted a much smaller size of ROI than CAM or Grad-CAM. In contrast, both CAM and Grad-CAM highlighted a much bigger region as class activation ROIs, and such ROIs included some part of the background, which had virtually no useful information. Table 3 also showed over 30% reduction of the ROI size, on average, by the proposed CRM. Note that a higher temperature (red and yellow-colored) area on a heatmap represented the most discriminative and important ROI in the corresponding image for modality classification. Examples in Figure 5e,f show that the CRM successfully localized and highlighted an object surrounded by a white dotted circle and the optic disc area in fundus as the most discriminative ROI, respectively, while CAM and Grad-CAM failed to do so.

We verified the consistency and localization performance of our method, shown in Figure 6, by comparing the normal distribution curves derived from three mapping scores (normalized to the range [0,1]) for all images in each modality class. As expected, CAM and Grad-CAM generated very similar distributions of the mapping scores for all modality classes. A distribution of the CRM score was found to have substantially smaller mean and standard deviation (STD) values than those from CAM and Grad-CAM scores for the classes of (a) abdomen CT, (b) brain MRI, (c) cardiac abdomen ultrasound, (d) chest X-ray, and (f) retinal fundoscopy, implying that the ROI within an image defined by our CRM was consistently smaller and more relevant than other methods. The CRM distributions for the fluorescence microscopy and statistical graph classes, shown in Figure 6e,g, where most images contained random sized and shaped objects or patterns, showed a mean value smaller than and a STD similar to those in CAM and Grad-CAM distributions. Such distribution results underscored all visual and quantitative findings from Figure 4 and Figure 5 and Table 3.

For further verification, we applied CAM and our CRM to two other types of pre-trained DL models: Inception-v3 [31] and Xception [32], and compared their localization performances. Both DL models had a more complicated structure and deeper convolutional layers than VGG16, and they also showed a high performance in many transfer learning-based image classification tasks. We first removed several convolutional layers from each of these DL models to generate a higher spatial resolution of feature maps at the last convolutional layer before GAP layer to improve their localization ability, as mentioned in [19]. Specifically, we removed the layers after “mixed6” and “add_11” from Inception-v3 and Xception, respectively, resulting in 17 × 17 of feature maps for each DL model. We then fine-tuned these DL models using our training set after adding a convolutional layer, followed by a GAP and a dense layer.

Figure 7 and Figure 8 show some examples of CAM and CRM heatmaps generated from Inception-V3 and Xception-based DL models. We observed that the proposed CRM consistently performed better than CAM in noise removal—especially in reduction of the highlighted background—and discriminative object detection and localization, similar to our findings in Figure 5. Thus, this further validated our method of generating a visual illustration where the DL model focused in a given image for the correct classification of its modality. 

Lastly, we also generated a “class-level” ROI using the proposed “average_CRM”. It represented the image region of greatest attention by the CNN for the correct prediction of images belonging to a particular class. Figure 9 presents the corresponding heatmaps of the “average_CRM” for each modality class. Here, “class-level” ROI was defined as the area having the “average_CRM” score above 70% of the maximum score and was visualized by enclosing it with a bounding box. It can be seen clearly that the heatmap of each modality class had a different size, shape, and location of “class-level” ROI. Such differences in the “class-level” ROI between modality classes would serve as a visual explanation for our “multi-modality” CNN model’s high performance (over 98%) in classifying image modalities. In the case of the fluorescence microscopy class, its corresponding “class-level” ROI was found to be a big rectangle shape covering almost the entire heatmap image, as shown in Figure 9e. This was very likely due to a very low degree of “intra-class” similarity between images.

In future research, we plan to employ and evaluate other state of the art DL architectures toward self-discovering hierarchical feature representations from the medical imaging modalities. The optimal DL model would be used to more reliably and effectively localize and visualize the discriminative ROIs in the medical modality images. We will also develop a generalized version of our CRM to visualize the internal representations of not only the last convolutional layer but also any other intermediate convolutional layers for a further and deeper understanding and interpretation of a DL model.

## 5. Conclusions

Image modality classification has gained immense significance as a first step to using the enormous medical image repositories generated through different imaging modalities for researching and developing more reliable computer-aided disease screening and diagnostic systems. Deep learning (DL) techniques, especially Convolutional Neural Networks (CNN), have made a breakthrough in extracting and analyzing information pertaining to the imaging modalities with minimal human intervention, and they have been reported to outperform other conventional machine learning methods. However, none of the DL-based modality classification studies attempted to interpret the internal representations and prediction results of their DL models. 

In this paper, we have proposed a novel method called “Class-selective Relevance Mapping” (CRM) for localizing and highlighting regions in a given image that are the most discriminative in classifying medical image modalities in order to visually explain the learned behavior of CNN-based DL models. Based on a linear sum of incremental MSE calculated from all output nodes, the CRM measures the importance and contribution of each spatial element in the feature maps produced from the last convolution layer to maximizing the difference between the prediction score at the output node for the desired class and those at the other output nodes.

In our experiments, we first developed a high-performing “multi-modality” CNN-based DL model using the pretrained VGG16 for classifying seven different types of image modalities, and then we analyzed its prediction behavior for a test image by visually localizing a discriminative ROI within the image using the proposed CRM. Experimental results show that our CRM consistently highlights a specific area of an image that is common in shape, pattern, or location among the other images within the same class, but they are distinct or different from images in other modality classes. We also visually and quantitatively evaluate the CRM performance as being substantially better than other existing localization methods in detecting a discriminative object or area within an image in terms of consistency, accuracy, and the size (number of pixels) of ROI highlighted. Such superior localization ability of our CRM is further validated by applying it to other types of DL models. Lastly, a different size, shape, and location of “class-level” ROI generated using the “average_CRM” visually explains over 98% of high performance with respect to all metrics our CNN model achieved in classifying image modalities.

## Figures and Tables

**Figure 1 diagnostics-09-00038-f001:**
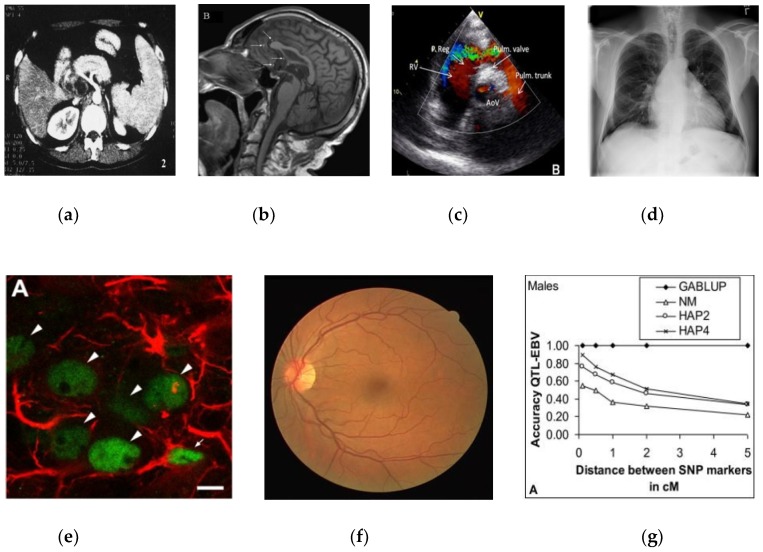
A sample image of each modality category: (**a**) abdomen CT, (**b**) brain MRI, (**c**) cardiac abdomen ultrasound, (**d**) chest X-ray, (**e**) fluorescence microscopy, (**f**) retinal fundoscopy, and (**g**) statistical graphs.

**Figure 2 diagnostics-09-00038-f002:**
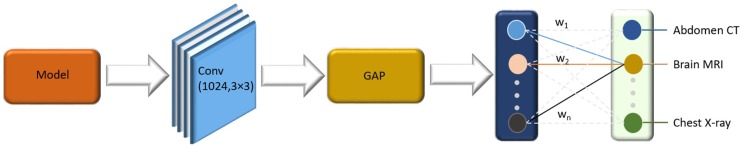
VGG16 network truncated at the deepest convolutional layer and added with a convolutional, global average pooling (GAP), and dense layer.

**Figure 3 diagnostics-09-00038-f003:**
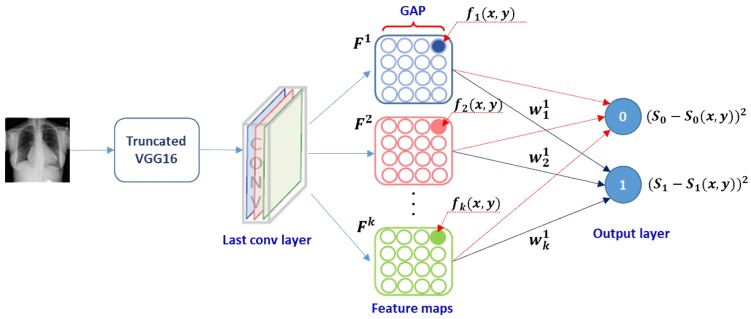
Calculation of class-selective relevance mapping (CRM) from a convolutional neural network (CNN)-based deep learning (DL) model.

**Figure 4 diagnostics-09-00038-f004:**
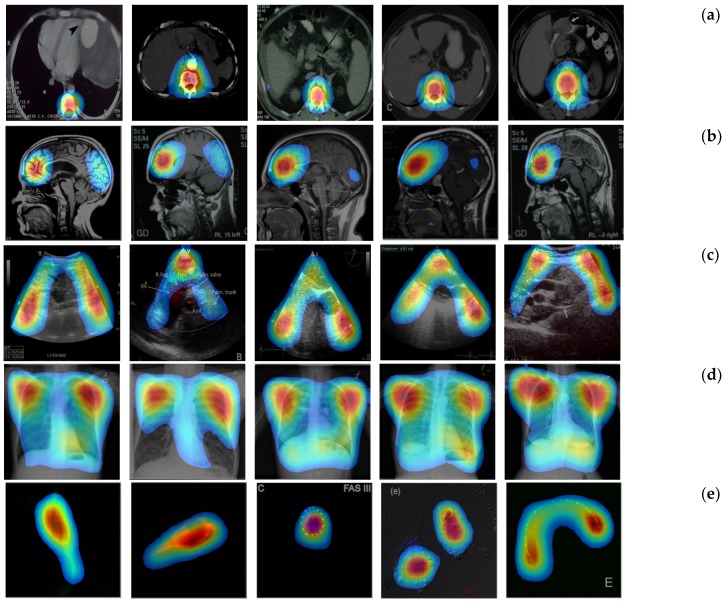
Heatmaps reflecting the proposed CRM for (**a**) abdomen CT, (**b**) brain MRI, (**c**) cardiac abdomen ultrasound, (**d**) chest X-ray, (**e**) fluorescence microscopy, (**f**) retinal fundoscopy, and (**g**) statistical graph images.

**Figure 5 diagnostics-09-00038-f005:**
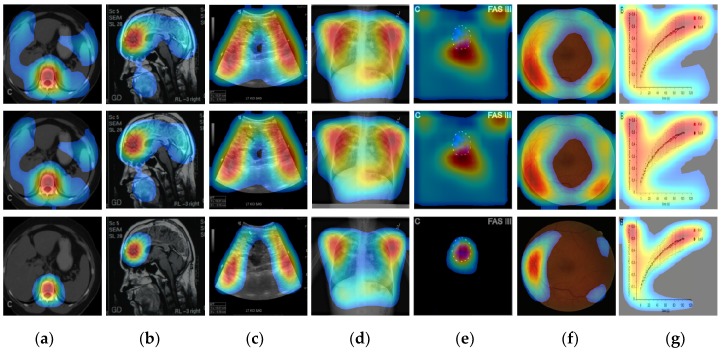
Examples of CAM (top row), Grad-CAM (middle row), and the proposed CRM (bottom row) for (**a**) abdomen CT, (**b**) brain MRI, (**c**) cardiac abdomen ultrasound, (**d**) chest X-ray, (**e**) fluorescence microscopy, (**f**) retinal fundoscopy, and (**g**) statistical graph images.

**Figure 6 diagnostics-09-00038-f006:**
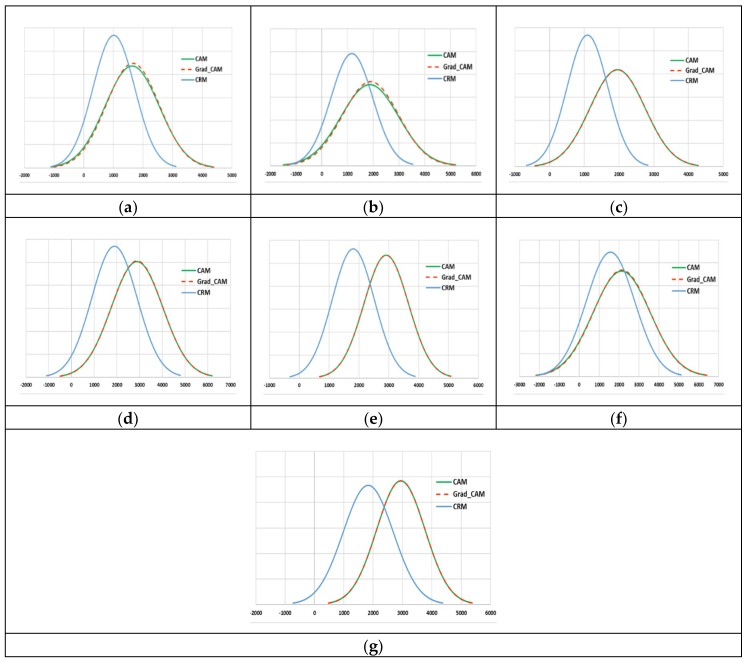
Normal distribution curves of three mapping scores for the classes of (**a**) abdomen CT, (**b**) brain MRI, (**c**) cardiac abdomen ultrasound, (**d**) chest X-ray, (**e**) fluorescence microscopy, (**f**) retinal fundoscopy, and (**g**) statistical graph images.

**Figure 7 diagnostics-09-00038-f007:**
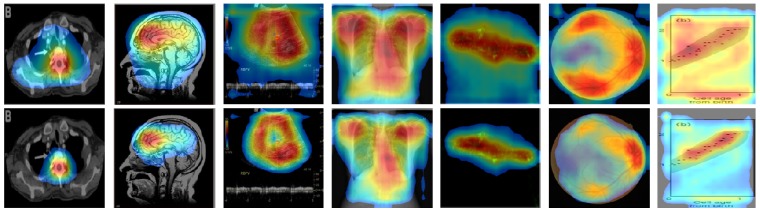
Examples of CAM (top row) and the proposed CRM (bottom row) heatmaps generated from InceptionV3-based “multi-modality” DL model.

**Figure 8 diagnostics-09-00038-f008:**
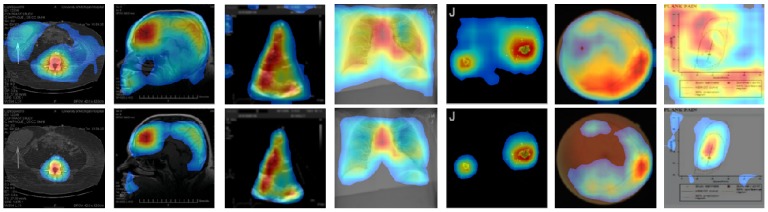
Examples of CAM (top row) and the proposed CRM (bottom row) heatmaps generated from Xception-based “multi-modality” DL model.

**Figure 9 diagnostics-09-00038-f009:**
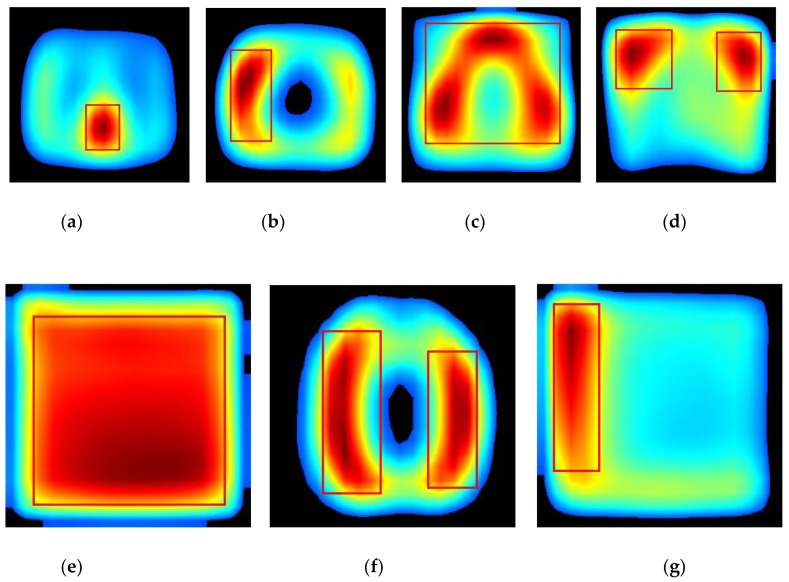
“Class-level” ROI for the classes of (**a**) abdomen CT, (**b**) brain MRI, (**c**) cardiac abdomen ultrasound, (**d**) chest X-ray, (**e**) fluorescence microscopy, (**f**) retinal fundoscopy, and (**g**) statistical graph.

**Table 1 diagnostics-09-00038-t001:** Data distribution across the image modalities.

Category	Samples	Training	Validation	Testing	File Type	Bit-Depth
Abdominal CT	6000	5000	500	500	JPG	8-bit
Brain MRI	6000	5000	500	500	JPG	8-bit
Chest X-ray	6000	5000	500	500	JPG	8-bit
Cardiac abdomen ultrasound	6000	5000	500	500	JPG	8-bit
Fluorescence microscopy	6000	5000	500	500	JPG	8-bit
Retinal fundoscopy	6000	5000	500	500	JPG	8-bit
Statistical graphs	6000	5000	500	500	JPG	8-bit

**Table 2 diagnostics-09-00038-t002:** Performance of VGG16-based “multi-modality” CNN model.

Model	Accuracy	AUC	Recall	Precision	F1-score	MCC
VGG16	0.98	0.994	0.98	0.981	0.98	0.986

**Table 3 diagnostics-09-00038-t003:** Average number and ratio (shown in parentheses) of pixels in 300 × 300 size of images in each modality class highlighted as region of interest (ROI) by each visualization method.

Methods	Abdomen CT	Brain MRI	Cardiac Abdomen Ultrasound	Chest X-ray	Fluorescence Microscopy	Retinal Fundoscopy	Statistical Graphs
CAM	49,477 (55.0)	54,894 (61.0)	56,972 (63.3)	76,488 (85.0)	79,900 (88.8)	58,514 (65.0)	82,444 (91.6)
Grad-CAM	49,478 (55.0)	54,896 (61.0)	56,973 (63.3)	76,488 (85.0)	79,901 (88.8)	58,515 (65.0)	82,445 (91.6)
CRM	26,596 (29.6)	32,298 (35.9)	28,966 (32.2)	57,363 (63.7)	52,448 (58.3)	43,334 (48.1)	52,932 (58.8)

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
