# Peer review of "Visual Interpretation of Convolutional Neural Network Predictions in Classifying Medical Image Modalities"

_diagnostics, 2019, doi:10.3390/diagnostics9020038_

Reviewer 1 Report

This manuscript focuses on visual interpretation for medical image modalities by using CNN. There are some issues in it.
1.According to Fig. 5, localization ability for ROI seems to be determined by the threshold values of the three methods. It is hard to confirm which method is the best. Although CRM is the best for Fig. 5(e) because it matches the dash circle, I don't think it always performs most precisely for the other cases.
2. The phenomena in Fig. 6 don't confirm CRM performs better than CAM and Grad-CAM do. All the data used by CNN are digital, so the data can be modulated easily. When the parameters of CAM and Grad-CAM are modulated, their normal distributions can become more narrow and their mean values can become smaller.
3. CRM is claimed to perform better than CAM and Grad-CAM according to Figs. 7 and 8. What are the evidences?
4. Narrower ROIs are not better than  wider ROIs always. Too narrow areas may lose some important information.
In conclusion, this manuscript must be revised.

Author Response

1. All heatmap images shown in our paper were generated by normalizing the mapping score to the range [0, 255] and displaying the value above 20% of the max score of each visualization method, as CAM [19] did. We can see from Fig. 5 that both CAM and Grad-CAM highlight a much bigger area within the images as their ROI. However such ROIs even include some part of the background, which has virtually no useful information. In contrast, CRM consistently has a much smaller size of ROI in which the background is minimally included. Therefore, our CRM can be considered to have a significant noise removal effect on heatmaps. Note that a higher temperature (red and yellow-colored) area on a heatmap represents the most discriminative and important ROI in the corresponding image for modality classification. Examples in Fig. 5 (e) and (f) show that the CRM successfully localizes and highlights an object surrounded by a white dotted circle and the optic disc area in fundus as the most discriminative ROI, respectively, while CAM and Grad-CAM fail to do so. We have updated the description of Fig. 5 to more clearly explain the noise removal and localization ability of our CRM.

2. CAM/Grad-CAM and CRM are based on a weighted sum of the activation of the feature maps at the deepest convolution layer and a linear sum of mean squared errors (MSE) at the output layer, respectively. Thus, given a deep learning (DL) model and an input image, there is no parameter that can affect their scores. In addition, we calculated a statistical distribution of the mapping score of each visualization method using the image dataset pooled from well-known and publicly available multiple medical image collections to ensure the accuracy and objectivity of our experiments.

3. Like Fig. 5, Figs. 7 and 8 also show that the proposed CRM demonstrates a superior performance in noise removal — especially in reduction of the highlighted background area — and enhanced object detection and localization ability compared to CAM. We have updated the descriptions of Figs. 7 and 8 to provide an intuitive explanation of these aspects.

4. Through a series of experiments, we visually and quantitatively evaluated and validated the CRM performance as being substantially better than other existing visualization methods in detecting the most discriminative ROI within an image in terms of not only the size of ROI but also consistency, accuracy, and noise removal ability.

Reviewer 2 Report

This study proposed a novel method called “Class-selective Relevance Mapping” (CRM) for localizing and visualizing discriminative regions of interest (ROI) within a medical image. Such visualizations offer improved explanation of the convolutional neural network (CNN)-based DL model predictions. We demonstrate CRM effectiveness in classifying medical imaging modalities toward automatically labeling them for  visual information retrieval applications.

(1)   Table 1, why images are stored in jpeg? Jpeg may be lossy compression.

(2)   Why choose VGG16? Did you consider VGG19 or googlenet?

(3)   Did you use transfer learning?

(4)   What is the effect of “class-selective relevance map”? Did you validate its effect through experiment?

(5)   How do you determine the structure of your neural network?

(6)   Why not use data augmentation?

(7)   Several papers could be discussed, see “Image based fruit category classification by 13-layer deep convolutional neural network and data augmentation” and “Alcoholism identification via convolutional neural network based on parametric ReLU, dropout, and batch normalization

(8)   What methods can be used to further improve the performance? You can discuss them in the future work.

Author Response

1. The dataset for training and testing of our DL models was pooled from multiple resources. Most of these images are available at the source in JPEG format. For this reason, we used JPEG-formatted images.

2. Our study aims to propose a novel method for localizing and visualizing a region of interest within a medical image that is considered to be the most discriminatively important in order to explain the predictions of CNN-based DL models toward the challenge of classifying medical imaging modalities. We used VGG16 since it performed very well in the 2014 ILSVRC, scored first place on the image localization task and second place on the image classification task. The model is also shown to demonstrate a strong ability to generalize to images outside the ImageNet dataset via transfer learning. For further verification of our method, we also employed other types of DL models: Inception-V3 and Xception. Since VGG19 and GoogLeNet have a similar structure or concept to the DL models used in our study, the proposed visualization method can be surely applied to, and is highly expected to have similar localization results from them.

3. Yes, we did. We first truncated the pretrained VGG16 at the deepest convolutional layer, and then added a convolutional layer with 512 filters, each of 3×3 spatial dimension, followed by a global average pooling and a single dense layer. The model was trained with small weight updates to learn the hierarchical feature representations in modality images and resulted in end-to-end feature extraction and classification.

4. We proposed “class-selective relevance mapping” (CRM) to localize and highlight regions in a given image that are the most important and discriminative in classifying medical image modalities, thereby visually explaining the learned behavior of CNN-based DL models. Our CRM is based on a linear sum of incremental MSE calculated from all output nodes to measure both positive and negative contributions of each spatial element in the feature maps produced from the last convolution layer, leading to correct classification of an input image. Therefore, the resulting ROI is intrinsically class-discriminative. In contrast, CAM and Grad-CAM rely on only the prediction score from the output node for a particular class to which a given input image belongs. Through a series of experiments, we visually and quantitatively evaluated and validated the CRM performance as being substantially better than other existing visualization methods in detecting a discriminative ROI within an image in terms of the size of ROI, consistency, accuracy, and noise removal ability.

5. A specific CNN architecture having global average pooled convolutional feature maps fed into the fully connected final output layer is required to apply three visualization methods (CAM, Grad-CAM, and our CRM) and evaluate their localization ability. Such CNN architecture was built based on the pretrained VGG16; we truncated the pretrained VGG16 at the deepest convolutional layer, and then added a convolutional layer, followed by a global average pooling and a single dense layer. The model was trained with small weight updates to learn the hierarchical feature representations in modality images. A randomized grid search was performed to find the optimal values for the model hyperparameters. The search ranges for the learning rate, momentum, and L2-regularization are set as [1e-5, 1e-1], [0.8, 0.99], and [1e-10, 1e-1] respectively. The optimal values for the learning rate, momentum, and L2-regularization were found to be 1e-4, 0.9, and 1e-6 respectively.

6. We did use data augmentation. The dataset is augmented with class-specific perturbations to make the model robust and reduce bias and generalization errors. These perturbations generate new samples for training without actually altering the visual characteristics of the images. Data is augmented with rotations and horizontal and vertical shifts in the ranges [-5, 5] and [-3, 3] respectively. Images are resampled to 300×300 pixel resolutions and mean-normalized to assist in faster convergence.

7. Many thanks for letting us know about these articles. The studies are referred to in our revised manuscript.

8. As a future work, we plan to employ and evaluate other state of the art DL architectures toward self-discovering hierarchical feature representations from the medical imaging modalities. The optimal DL model would be used to more reliably and effectively localize and visualize the discriminative ROIs in the medical modality images. We will also develop a generalized version of our CRM to visualize the internal representations of not only the last convolutional layer but also any other intermediate convolutional layers for a further and deeper understanding and interpretation of a DL model. A brief description of our future work has been provided in the revised manuscript.

Reviewer 3 Report

This paper is well written and is technically sound. 

The maths seems correct. However the paper strength could be improved in the case of the Authors are able to compare the performance of the proposed CNN model with other reported from literature (e.g. considering techniques that have been applied to available datasets).

Author Response

Our study aims to propose a novel method called “Class-selective Relevance Mapping” (CRM) for localizing and visualizing a region of interest (ROI) within a given medical image that is the most important and discriminative in classifying medical image modalities in order to visually explain the learned behavior of CNN-based deep learning (DL) models. Through a series of experiments on three different types of DL models: VGG16, Inception-V3, and Xception-based, we visually and quantitatively evaluated and validated the CRM performance as being significantly better than other existing visualization methods (CAM and Grad-CAM) in detecting the most discriminative ROI within an image in terms of the size of ROI, consistency, accuracy, and noise removal ability. In future research, we plan to employ and evaluate other state of the art DL architectures toward self-discovering hierarchical feature representations from the medical imaging modalities. The optimal DL model would be used to more reliably and effectively localize and visualize the discriminative ROIs in the medical modality images.

We have updated the descriptions of Figs. 5, 7, and 8 to more clearly explain the evaluation results and provided a brief description of our future work in the revised manuscript.

Reviewer 4 Report

Authors investigated deep learning models such as convolutional neural networks (CNN) for predicting the medical images. Special focus was given on visual interpretation. This paper is inspired by increasing use of CNNs for development of reliable computer-aided detection, diagnosis and information retrieval methods. Interpretation of CNN model in medical scenarios is useful for ensuring rapid adoption and usage in real-world systems. Authors developed a novel approach known as “Class-selective Relevance Mapping” (CRM) for localizing and visualizing regions of interest within a medical image. These visuals help in explaining the model predictions from deep CNN on medical images. Experiments validated the efficacy of CRM in classifying medical imaging modalities. It would help in automatic labeling them for visual information retrieval applications.

The proposed method is based on linear sum of incremental mean squared errors (MSE) computed from output layer in CNN. Thus, it accounts for both positive and negative contributions from each spatial element in the feature maps at output layer. Output layer leads to accurate classification of the input image. Authors did comparative experiments using multi-modal CNN for classification of different image types and found the proposed method to be significantly better than state of the art approaches.

Authors visualized the class-specific maps by averaging the scores of images in each class and explained the visual explanation through their different size, shape, and location for the proposed CNN model. This model achieved over 98% performance on a dataset derived from public images.

Paper is well explained and supported with experimental results. Results validated the proposed method on studied task and hence acceptable paper quality.

Author Response

The authors would like to thank the reviewers for their precious time and invaluable comments. We have carefully addressed all the comments.

Round  2

Reviewer 1 Report

The revised manuscript is ready to be published.

Reviewer 2 Report

Accept